# MetaRepair: Learning to Repair Deep Neural Networks from Repairing Experiences

## ABSTRACT

Repairing deep neural networks (DNNs) to maintain its performance during deployment presents significant challenges due to the potential occurrence of unknown but common environmental corruptions. Most existing DNN repair methods only focus on repairing DNN for each corruption separately, lacking the ability of generalizing to the myriad corruptions from the ever-changing deploying environment. In this work, we propose to repair DNN from a novel perspective, *i.e.* Learning to Repair (L2R), where the repairing of target DNN is realized as a general learning-to-learn, *a.k.a.* meta-learning, process. In specific, observing different corruptions are correlated on their data distributions, we propose to utilize previous DNN repair experiences as tasks for meta-learning how to repair the target corruption. With the meta-learning from different tasks, L2R learns a meta-knowledge that summarizes how the DNN is repaired under various environmental corruptions. The meta-knowledge essentially serves as a general repairing prior which enables the DNN quickly adapt to unknown corruptions, thus making our method generalizable to different type of corruptions. Practically, L2R benefits DNN repair with a general pipeline yet tailoring meta-learning for repairing DNN is not trivial. By re-designing the meta-learning components under DNN repair context, we further instantiate the proposed L2R strategy into a concrete model named MetaRepair with pragmatic assumption of experience availability. We conduct comprehensive experiments on the corrupted CIFAR-10 and *tiny*-ImageNet by applying MetaRepair to repair DenseNet, ConvNeXt and VAN. The experimental results confirmed the superior repairing and generalization capability of our proposed L2R strategy under various environmental corruptions.

## CCS CONCEPTS

• **Computing methodologies** → **Transfer learning**.

## KEYWORDS

DNN repair, meta-learning, model generalization, learning with noise.

## 1 INTRODUCTION

Deep neural networks (DNNs) have shown discernible achievements in a wide range of applications [19], ranging from image recognition [17, 35], machine translation [13, 34], to safety-critical domains like medical diagnosis [23] and self-driving [3]. In spite of the excellent performance DNNs have demonstrated, applying a well pretrained DNN for practical application is still challenging due to the various discrepancy between experimental and deploying environments. One of the most common obstacles is the input corruption [41], where the pretrained DNN has to serve a totally different source of inputs that oftentimes fail DNN for correct function. In an effort to maintain the performance of DNN during model deployment, many approaches have been proposed to repair DNN from both software engineering [5, 30–32, 39] and machine learning [26, 37, 41, 42] perspectives.

Conventionally, as shown in Fig. 1 (a), existing repairing approaches simply utilize the collected few failure examples to repair DNN for the target deploying environment [26, 30]. However, such pipeline faces a crucial issue [26]: ***the DNN is repaired only for this very target corruption and cannot generalize to other situations after repairing***. During practical DNN deploying process, the deploying environment keeps changing which makes there are numerous corruptions need to be repaired. It is unrealistic and inefficient to carry out the repairing pipeline for each corruption one by one. In fact, the environmental corruptions can be conceptually categorized by their characteristics [9], *e.g.* blur, noise, weather, *etc.*. In consequence, a desired DNN repair method should be at least generalize to corruptions of the same category. Nevertheless, generalization of DNN, especially when data is scarce, is a long-standing tough task in the research of machine learning [43].

Although the discussion on generalizable DNN repairing is scarce, there are plenty of established generalization studies on other topics [20, 24, 38, 44]. For example, in the research of few-shot learning [27], meta-learning has demonstrated its superior generalization capability by transform the learning objective into a learning-to-learn problem [4]. In this work, inspired by the great generalizability that meta-learning has demonstrated in other problems [21], we introduce meta-learning into the problem of DNN repair. As shown in Fig. 1 (b), we propose the Learning to Repair (L2R) strategy where the DNN repair is formalized into a learning-to-learn problem. Our aim with L2R is to learn the repairing capability by conducting meta-learning among the repairing experience of different corruptions. Inside the meta-learning, L2R first meta-train the DNN on collected failures of different corruptions for learning a meta-knowledge which essentially summarizes how the DNN is updated and repaired under different corruptions. Later given a target corruption needs to be repaired, the meta-knowledge serves as instructions to repair the DNN for target corruption during meta-test. Basically, L2R is the learning and application of the repairing ability rather than the repairing of specific corruption, which makes the pipeline corruption-independent. Therefore, our strategy is expected to be generalizable for repairing DNN under various corruptions.

Practically, our L2R benefits DNN repair with a generalizable pipeline but tailoring meta-learning for DNN repair is not trivial.

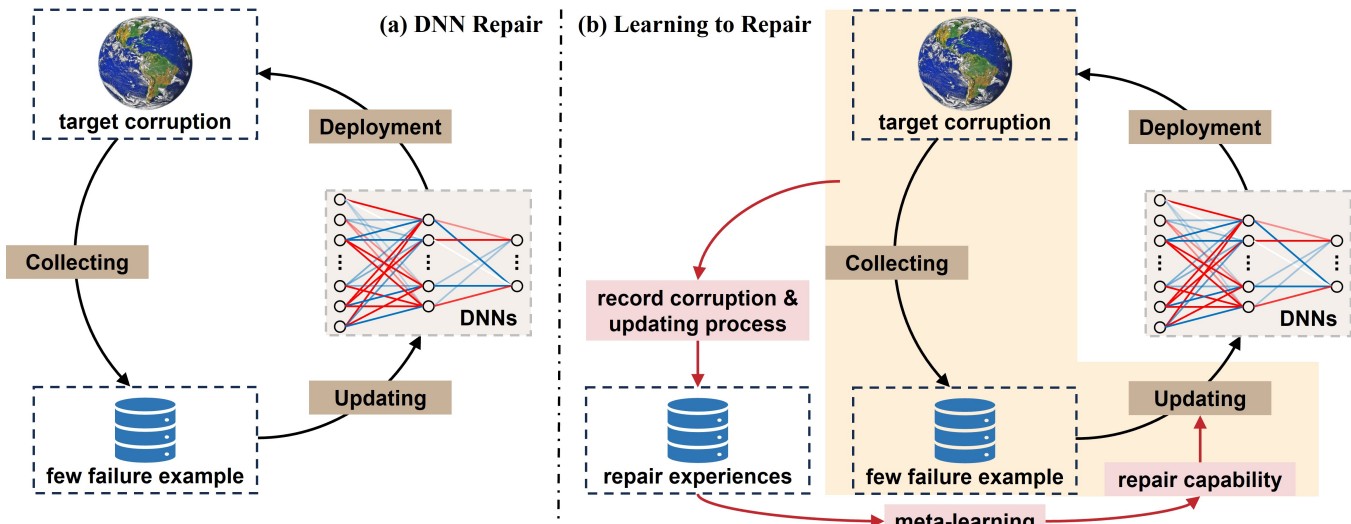

Figure 1: Left: the conventional DNN Repairing pipeline. Right: our proposed Learning to Repair strategy.

In specific, there are three essential components needs to be considered when formalizing DNN repair as a meta-learning problem. ❶ The task distribution on which meta-learning is carried out, ❷ the meta-learner and base-learner as well as the meta-knowledge that represents repairing capability and ❸ the meta-training and testing procedure designed under DNN repair context. Therefore, we further provide a concrete instantiation of L2R named MetaRepair to verify the proposed repairing strategy. Fundamentally, our work benefits DNN repair research with the following contributions.

- In this work, we provide a novel perspective for DNN repair. We redefine the conventional DNN repairing into a meta-learning process and elaborate the new challenges as well as the model, dataset and learning setup.
- We propose a novel Learning to Repair (L2R) strategy to repair DNN with generalization. Our L2R learns how to repair DNN under different corruptions with meta-learning rather than only repairing for specific corruption, which is desirable for practical DNN deployment.
- We show that the proposed L2R can be realized with great flexibility by constructing the MetaRepair model which is one of various realizations of L2R strategy.
- With extensive experiments, the results of repairing three modern DNNs on corrupted CIFAR-10 and *tiny*-ImageNet demonstrate the proposed L2R strategy has superior repairing and generalization capabilities.

## 2 RELATED WORK

DNN repair aims to mitigate the performance degradation at the model deployment stage. Existing DNN repair approaches can be roughly categorized into following two types.

Researchers from the software engineering community commonly take DNN as a kind of software and view the performance degradation as a kind of software bugs [5, 32, 39]. To localize and rectify the buggy behaviors of deployed DNN, they investigate the DNN construction like activation function [5], model weights [39],

or network patches [32] etc. to find which part of the DNN should be adjusted. However, the DNN is usually a black box for the consideration of safety or commercial reasons [1] during practical deployment which makes it hard for structure analysis. Moreover, repairing DNN with structure investigation potentially leads to sub-optimal solutions [14] since the target deploying environment is keeping changing.

Another kind of approach consider the repairing problem as further optimization of the DNN, *i.e.* repairing-by-optimization method, under specific corruption with the supervision of a few collected failure examples [26, 41, 42], which is familiar to researchers with machine learning backgrounds as fine-tuning. Nevertheless, correcting DNN misbehavior by further optimization is not non-trivial due to the hard of training data accessing [41] and ease of over-fitting [42], both of which are prone of introducing new misbehaviors. One way of realizing DNN repair by further optimization with limited failure cases and avoiding over fitting is to augment the data with cutting-edge generation techniques [41], but they are notoriously reported to be easy of mode collapsing [28] and incapable of accurate distribution matching [15].

In general, existing DNN repair methods either unsuitable for actual deploying process or lack the capability of generalizing to different situations. Therefore, we introduce meta-learning into DNN repair in order to realize DNN repair with promising generalization performance. As an established learning paradigm, meta-learning [21, 33] is designed to be effective in data-scarce and generalization-critical problems. Particularly, meta-learning is especially suitable for DNN repair considering it can be realized as model-agnostic [4]. It suggests that we are not design the repairing method for specific target environment which substantially makes meta-learning applicable to DNN repair problem.

## 3 PROBLEM STATEMENT

We first introduce the conventional definition of DNN repair and re-formalize it in Sec. 4 into a meta-learning process. For the main

notations utilized, we refer readers to the supplemental material for a comprehensive grasp.

Typically, a DNN $F_\theta(\cdot)$ parameterized with $\theta$ is learned with some training data $\mathcal{D}_{train}$ and evaluated on the corresponding testing data $\mathcal{D}_{test}$, where $\mathcal{D}_{train}/\mathcal{D}_{test} \sim \mathcal{P}_{clean}$. When deploying the pretrained DNN $F_\theta(\cdot)$ into a target real-world environment, we preconceive the data encountered also comes from $\mathcal{P}_{clean}$. However, such assumption is almost not the truth in reality [9] and it always fails the DNN for correct functioning [40]. In fact, disturbed by the environmental corruption, the data of the target deploying environment $\mathcal{D}_{corrupt}$ usually follows a different distribution $\mathcal{D}_{corrupt} \sim \mathcal{P}_{corrupt}$.

To maintain the DNN performance under the environmental corruptions, DNN repair [26, 30] is proposed to rectify DNN misbehaviour by utilizing a few failure cases $\mathcal{D}_{collect} \subset \mathcal{D}_{corrupt}$ collected from the target deploying environment. After repairing, the repaired DNN $F_{\hat\theta}(\cdot)$ is expected to perform correctly on $\mathcal{D}_{fail} \subset \mathcal{D}_{corrupt}$, where $\mathcal{D}_{fail} \cap \mathcal{D}_{collect} = \emptyset$. Mathematically, the above DNN repair process can be formulated as

$$\max \mathbb{E}_{(X,y)\in\mathcal{D}_{fail}} \mathcal{M}(F_{\hat\theta}(X), y) \tag{1}$$

$$s.t.$$

$$\hat\theta = \arg\min_\theta \mathbb{E}_{(X,y)\in\mathcal{D}_{collect}} \mathcal{L}(F_\theta(X), y) \tag{2}$$

where $\mathcal{M}$ is the performance metric adopted and $\mathcal{L}$ is the loss for DNN optimization. However, as the notation indicated above, this objective is only the repairing for one specific corruption $\mathcal{P}_{corrupt}$. Without consideration of the correlation between corruptions, the DNN repaired by such strategy can only be applicable for the target environment and incapable of generalizing to other similar deploying environments.

## 4 METHODOLOGY

To the end of repairing DNN with generalizability, we propose to formalize DNN repair into a meta-learning process which can be elaborated with following two aspects.

- We first redefine DNN repair into a meta-learning problem with Learning to Repair (L2R) strategy in Sec. 4.1,
- then in Sec. 4.2, we consider the implementation of L2R and design the MetaRepair model for experiments.

### 4.1 Learning to Repair

As illustrated in Fig. 1 (b), the general idea of L2R is that the ability of repairing DNN for target corruption is learned from the experiences of repairing DNN for other corruptions. Essentially, such a learning to repair capability is acquired with a basic hypothesis:

**Hypothesis** *The data distribution of K different deploying environments $\{\mathcal{P}_{corrupt}^k\}_{k=1}^K$ are correlated in a sense that similar deploying environments match better on their data distributions.*

This hypothesis has been implicitly applied to various noise-related machine learning researches [43], and we also demonstrate its genuineness with experiments in Sec. 5.3. Fundamentally, the hypothesis provides us a premise for realizing generalizable DNN repairing. Making use of the repairing experience of correlated corruptions, we can expect the repairing of target corruption will be facilitated when they share similar data distribution.

Practically, given $K$ deploying environments $\{\mathcal{P}_{corrupt}^k\}_{k=1}^K$ and the corrupted data $\{\mathcal{D}_{corrupt}^k = (\mathcal{D}_{collect}^k, \mathcal{D}_{fail}^k)\}_{k=1}^K$, we setup the DNN repairing process for each corruption

$$\hat\theta_k = \arg\min_\theta \mathbb{E}_{(X,y)\in\mathcal{D}_{collect}^k} \mathcal{L}(F_\theta(X), y) \tag{3}$$

where $\mathcal{L}$ refers to the loss of DNN repairing. To facilitate the repairing of target corruption with these $K$ repairing experiences, we take them as meta-learning tasks and meta-train the DNN to learn how it should be repaired under different corruptions. Accordingly, we further setup a meta-learning objective which can be formulated as

$$\xi_{meta} = \arg\min_{\xi_{\theta\to\hat\theta_k}} \mathbb{E}_{(X,y)\in\mathcal{D}_{fail}^k} \mathcal{L}_{meta}(F_{\hat\theta_k}(X), y, \xi_{\theta\to\hat\theta_k}) \tag{4}$$

where $\xi$ is the meta-knowledge that represents the repairing capability, $\mathcal{L}_{meta}$ is the meta-training loss and $\theta \to \hat\theta_k$ summarizes the repairing process of Eq. 3.

Intuitively, Eq. 4 learns an optimal representation of repairing ability by evaluating the repaired DNN on the failure data of different corruptions. Following the hypothesis, such repairing capability $\xi_{meta}$ is also applicable to the target corruption when it has correlation to the $K$ repairing experiences. Therefore, the repairing of the target deploying environment $\mathcal{P}_{corrupt}^t$ is expected to be facilitated by further applying $\xi_{meta}$ for meta-testing

$$\hat\theta = \arg\min_\theta \mathbb{E}_{(X,y)\in\mathcal{D}_{collect}^t} \mathcal{L}(F_\theta(X), y, \xi_{meta}) \tag{5}$$

where $\xi_{meta}$ performs as a repairing prior for Eq. 3 to facilitate the repairing process. Note the final performance is calculated with

$$\mathbb{E}_{(X,y)\in\mathcal{D}_{fail}^t} \mathcal{M}(F_{\hat\theta}(X), y) \tag{6}$$

and $\mathcal{D}_{corrupt}^t = (\mathcal{D}_{collect}^t, \mathcal{D}_{fail}^t)$ are the collected and failure examples of the target deploying environment.

### 4.2 MetaRepair

We now further design a model dubbed MetaRepair by fleshing L2R out with consideration of critical meta-learning components under DNN repair context. Briefly, ❶ we first clarify the challenges and our designing of meta-learning task distribution for DNN repair. ❷ Then we setup the base- & meta-learner and meta-knowledge within L2R strategy. ❸ Finally, all the components are integrated into a comprehensive meta-learning procedure.

**Task Distribution** Traditionally, meta-learning is commonly formalized into the few-shot learning paradigm [4] where the tasks are assembled as class-level learning problems. For example [33], a few-shot learning task $\mathcal{T} = \{D_{train}, D_{test}, \mathcal{L}_\mathcal{T}\}$ consists of the training and testing data as well as a task-specific loss, where the samples composed of $D_{train}$ and $D_{test}$ are of the same class. During the meta-learning, the tasks are uniformly sampled over classes and the model is meta-trained and -tested with those tasks in an episodic manner [22]. ***Different from conventional meta-learning scenario, our L2R conducts meta-learning among the repairing experiences of different corruptions rather than classes. As a result, the tasks in our L2R are defined as corruption-level learning problems.***

Specifically, given the data $\mathcal{D}_{corrupt} = (\mathcal{D}_{collect}, \mathcal{D}_{fail})$ of specific corruption, we use the same format $\mathcal{T} = \{D_{train}, D_{test}, \mathcal{L}_\mathcal{T}\}$

to represent a corruption-level task where $D_{train} \subset \mathcal{D}_{collect}$ and $D_{test} \subset \mathcal{D}_{fail}$. The main difference is that $D_{train}$ and $D_{test}$ are of the same corruption rather than the same class. However, such task construction faces two critical issues. ❶ The classes composed of $D_{train}$ and $D_{test}$ may be totally disjoint, which is catastrophic for adaptation during meta-learning. Moreover, ❷ different tasks have different correlations to the target corruption according to the hypothesis in Sec. 4.1, which indicates the uniform sampling of tasks is inappropriate for DNN repair problem.

To tailor the meta-learning task for DNN repair, we propose to design it as semantic-cover and confidence-aware task to address the above issues. In general, the task for L2R is set as $\mathcal{T} = \{D_{train}, D_{test}, \mathcal{L}_{\mathcal{T}}, p_{corrupt}\}$, where the examples in $D_{train}$ now cover all classes and $p_{corrupt}$ is the corruption-wise probability for task sampling. Particularly, we choose Fréchet Inception Distance (FID) [11] as the empirical correlation measurement for corruptions which is given by

$$q_{corrupt}^{kt} = ||\mu_{corrupt}^k - \mu_{corrupt}^t||_2^2 +$$
$$tr(\Sigma_{corrupt}^k + \Sigma_{corrupt}^t$$
$$- \sqrt{2(\Sigma_{corrupt}^k \Sigma_{corrupt}^t)}) \quad (7)$$

where $\mathcal{N}(\mu_{corrupt}, \Sigma_{corrupt})$ is the normal distribution estimated with Inception v3 [36] feature of the collected examples. $k, t$ refer to the $k$-th and target corruption respectively.

Note that the more similar two deploying situations the smaller the computed FID score, where the same corruption has a FID score of 0. To apply it as confidence of repairing experience, we adopt the exponential function for converting the score into probability

$$p_{corrupt}^{kt} = \frac{e^{-\alpha * (q_{corrupt}^{kt} - \beta)}}{\sum_{k,t} e^{-\alpha * (q_{corrupt}^{kt} - \beta)}} \quad (8)$$

where $\alpha, \beta$ are threshold hyper-parameters of FID score.

**Base- & Meta-learner and Meta-knowledge** While the meta-learning task for DNN repair requires careful designing, we focus on the designing of repairing strategy rather than the tricky learning algorithm. Thus, we adopt the well-known MAML [4] algorithm to train our L2R. Within MAML algorithm, the base- and meta-learner are basically the same model which is exactly the DNN under repairing in our situation. Such designing makes the learning algorithm model-agnostic which is desirable for repairing different DNNs during deployment.

Simultaneously, the meta-knowledge in Eq. 4 is set to the parameters of DNN itself according to MAML. Although there are various choices to summarize the repairing process, *e.g.* learning rate [29], loss function [7] or manual hyper-parameters [2], the DNN itself is the most straightforward solution to both summarize the learning process and represent the repairing capability. In summary, the optimization of meta-knowledge, *i.e.* Eq. 4, now formulated as

$$\hat{\theta}_{meta} = \arg\min_{\theta \to \hat{\theta}_k} \mathbb{E}_{(X,y) \in \mathcal{D}_{fail}^k} \mathcal{L}_{meta}(F_{\hat{\theta}_k}(X), y, \theta \to \hat{\theta}_k) \quad (9)$$

where the $\mathcal{L}_{meta}$ is actually set to the $\mathcal{L}$ of Eq. 3 for simplicity during the experiments.

**Meta-learning Procedure** In the original MAML algorithm [4], the meta-learning is conducted in an episodic manner where each

episode composes of several independent tasks. During meta-testing, the base-learner is adapted and evaluated individually on each task. However, such meta-testing procedure is unsuitable for DNN repair where the testing tasks are related since they comes from the same corruption. Therefore, we apply episodic learning with essential modifications to suit DNN repair problem. **Specifically, as detailed in Algorithm L17-L21, we set the updates of base-learner during meta-test to be accumulated on the tasks in an episode.** This ensures the base-learner not only takes advantage of the repairing capability meta-learned on other corruptions, but also the data itself of target corruption. Overall, the proposed algorithm dubbed MetaRepair is detailed as follows.

---

**Algorithm 1:** MetaRepair

**Input** : $\{\mathcal{D}_{corrupt}^k = (\mathcal{D}_{collect}^k, \mathcal{D}_{fail}^k)\}_{k=1}^K$,
$\mathcal{D}_{corrupt}^t = (\mathcal{D}_{collect}^t, \mathcal{D}_{fail}^t)$, $F_\theta(\cdot)$, $\mathcal{L}$

1 compute $p_{corrupt}^{kt}$ with Eq. 7 and Eq. 8
2 **while** *not done* **do**
3   **for** *episode* = $1 \to N$ **do**
    // $D_{train}^s \subset \mathcal{D}_{collect}^k$, $D_{test}^s \subset \mathcal{D}_{fail}^k$
4     sample an episode of meta-train tasks $\{\mathcal{T}_s\}_{s=1}^S$
5     $= \{(D_{train}^s, D_{test}^s, \mathcal{L}, p_{corrupt}^{kt})\}_{s=1}^S$
6     **for** *task* = $1 \to S$ **do**
7       $\hat{\theta}_s = \arg\min_\theta \mathbb{E}_{(X,y) \in D_{train}^s} \mathcal{L}(F_\theta(X), y)$
8       $\mathcal{L}_s = \mathbb{E}_{(X,y) \in D_{test}^s} \mathcal{L}(F_{\hat{\theta}_s}(X), y)$
9     **end**
10     $\hat{\theta}_{meta} = \arg\min_\theta \sum_{s=1}^S \mathcal{L}_s$
11     $\theta \leftarrow \hat{\theta}_{meta}$
12   **end**
13   **for** *episode* = $1 \to M$ **do**
    // $D_{train}^s \subset \mathcal{D}_{collect}^t$, $D_{test}^s \subset \mathcal{D}_{fail}^t$
14     sample an episode of meta-test tasks $\{\mathcal{T}_s\}_{s=1}^S$
15     $= \{(D_{train}^s, D_{test}^s, \mathcal{L})\}_{s=1}^S$
16     $\hat{\theta} \leftarrow \hat{\theta}_{meta}$
17     **for** *task* = $1 \to S$ **do**
18       $\hat{\theta}_s = \arg\min_{\hat{\theta}} \mathbb{E}_{(X,y) \in D_{train}^s} \mathcal{L}(F_{\hat{\theta}}(X), y)$
19       $\hat{\theta} \leftarrow \hat{\theta}_s$
20       $acc_s = \mathbb{E}_{(X,y) \in D_{test}^s} \mathcal{M}(F_{\hat{\theta}}(X), y)$
21     **end**
22   **end**
23   $acc = (\sum_{m=1}^M \sum_{s=1}^S acc_s)/(M \times S)$
24 **end**

---

## 5 EXPERIMENT

In this section, we conduct experiments to answer the following four research questions (RQ).

- **RQ1:** How MetaRepair performs compared with state-of-the-art (SOTA) DNN repairing approaches?
- **RQ2:** Does the Hypothesis in Sec. 4.1 holds for DNN repairing problem?

**Table 1: The repairing performance of the deployed DenseNet repaired by SOTA methods and MetaRepair on CIFAR-10.**

| Corruption / Method | Clean | CIFAR-10 | | | | | | |
|---|---|---|---|---|---|---|---|---|
| | | GN | ZM | FOG | BR | PIX | JPEG | ET |
| Apricot [42] | | 37.03 | 20.90 | 36.89 | 40.63 | 42.25 | 39.41 | 37.11 |
| AugMix [10] | | 48.18 | 63.92 | 48.37 | 50.90 | 65.08 | 54.71 | 55.29 |
| Arachne [30] | | 37.15 | 26.36 | 45.89 | 38.98 | 33.81 | 40.47 | 49.48 |
| SENSEI [6] | 95.69 | 21.24 | 10.31 | 14.34 | 17.17 | 17.39 | 18.78 | 14.33 |
| DeepRepair [41] | | 61.51 | 73.14 | 57.81 | 56.94 | 60.32 | 63.41 | 62.56 |
| ArchRepair [26] | | 62.16 | 70.81 | 58.06 | 61.17 | 60.58 | 58.80 | 62.71 |
| **MetaRepair (*ours*)** | | **73.51** | **89.45** | **59.15** | **65.78** | **62.84** | **70.23** | **65.55** |

**Table 2: The repairing performance of the deployed DenseNet repaired by SOTA methods and MetaRepair on *tiny*-ImageNet.**

| Corruption / Method | Clean | *tiny*-ImageNet | | | | | | |
|---|---|---|---|---|---|---|---|---|
| | | GN | ZM | FOG | BR | PIX | JPEG | ET |
| Apricot [42] | | 17.80 | 13.31 | 16.85 | 15.99 | 12.59 | 15.96 | 11.83 |
| AugMix [10] | | 18.64 | 13.96 | 16.87 | 16.51 | 15.07 | 16.25 | 12.92 |
| Arachne [30] | | 18.23 | 13.94 | 17.09 | 15.47 | 10.87 | 16.20 | 14.76 |
| SENSEI [6] | 50.69 | 16.95 | 13.42 | 16.24 | 15.49 | 11.80 | 16.73 | 12.90 |
| DeepRepair [41] | | 18.80 | 14.09 | 16.34 | 14.93 | 13.21 | 17.12 | 13.50 |
| ArchRepair [26] | | 18.85 | 13.94 | 16.96 | 16.94 | 13.69 | 15.39 | 13.77 |
| **MetaRepair (*ours*)** | | **27.19** | **18.14** | **19.83** | **25.32** | **21.77** | **26.51** | **17.88** |

- **RQ3:** Can the DNN generalize to different deploying environments after repair?
- **RQ4:** What are the contributions of different MetaRepair components?

## 5.1 Experimental Setup

We first summarize the corruptions, datasets, the repairing setup, DNNs and the baseline in our experiments. For more experimental details, please refer to the supplemental material.

*5.1.1 Corruptions.* To simulate different DNN deploying environments $\{\mathcal{P}^k_{corrupt}\}^K_{k=1}$, we apply the commonly used corruptions [9] to existing data as the corrupted data source $\{\mathcal{D}^k_{corrupt}\}^K_{k=1}$. Also, the severity is set as 5 different levels following [9] for each corruption. Specifically, there are 19 corruptions in total which can be categorized as follows:

- Noise: gaussian noise (GN), shot noise (SN), impulse noise (IN), speckle noise (SKN).
- Blur: defocus blur (DB), glass blur (GB), motion blur (MB), zoom blur (ZM), gaussian blur (GAB).
- Weather: snow (SW), frost (FR), fog (FOG), spatter (SP).
- Image property: brightness (BR), contrast (CT), saturate (SA).
- Digital error: elastic transform (ET), pixelate (PIX), jpeg compression (JPEG).

As all the corruptions can be adopted as the repairing target, we take gaussian noise, zoom blur, fog, brightness as the representatives of target corruptions to be repaired. As for digital error, we experiment all the three corruptions since they are unique to each other.

*5.1.2 Datasets.* We evaluate MetaRepair on CIFAR-10 [16] and *tiny*-ImageNet [18]. CIFAR-10 contains 60,000 images in total for 10 categories with the size of (64, 64), where 50,000 images are split as training and the remaining 10,000 images are utilized for testing. Similarly, *tiny*-ImageNet is a subset of ImageNet [17] dataset and contains 200 classes of image with the size of (64, 64). It composed of a training split with 100,000 images, a validation split and a testing split both of which contain 10,000 images.

*5.1.3 Repairing Setting.* In general, one corruption is setup as the target corruption for meta-testing and $K$ out of the rest 18 corruptions are utilized for meta-training in a round of experiment. During meta-training, the data source $\{\mathcal{D}^k_{corrupt} = (\mathcal{D}^k_{collect}, \mathcal{D}^k_{fail})\}^K_{k=1}$ is constructed by applying the $K$ meta-training corruptions to the training split $\mathcal{D}_{train}$ of the clean dataset. Similarly, the data source for meta-testing $\mathcal{D}^t_{corrupt} = (\mathcal{D}^t_{collect}, \mathcal{D}^t_{fail})$ is constructed by applying the meta-testing corruption to the clean testing split $\mathcal{D}_{test}$.

In more specific, for the $k$-th corruption of meta-training, we randomly select 1000 failures from $\mathcal{D}^k_{corrupt}$ as the $\mathcal{D}^k_{collect}$ and the rest of it is utilized as $\mathcal{D}^k_{fail}$. As for meta-testing, we follow the same procedure as DeepRepair [41] to filter out all the examples failed the pretrained DNN from $\mathcal{D}^t_{corrupt}$. Then, 1000 examples are further randomly selected from them as $\mathcal{D}^t_{collect}$ and the rest are assembled into $\mathcal{D}^t_{fail}$.

For both meta-training and meta-testing, the tasks are assembled following the same procedure as described in the task distribution part of Sec. 4.2 with the data source described above. More details of task construction can be found in the supplemental material.

*5.1.4 DNNs.* We adopt Densenet 121 [12], ConvNeXt base [25] and Visual Attention Network (VAN) base [8] as the DNNs for repairing experiments. For all three selected DNNs, we pretrained them on

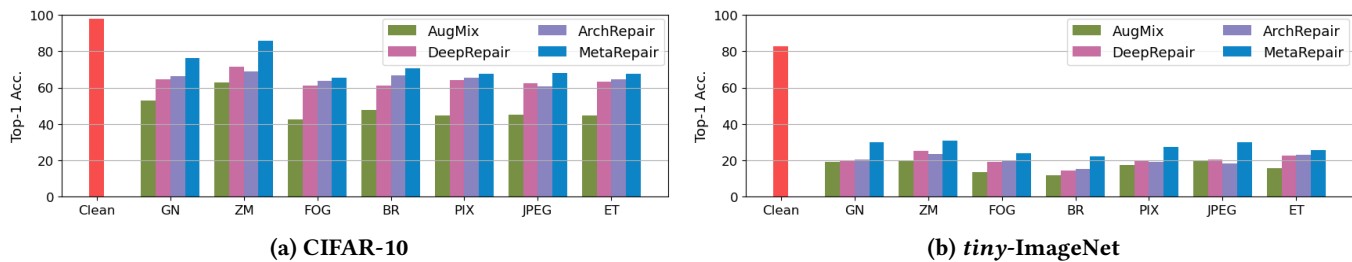

(a) CIFAR-10

(b) *tiny*-ImageNet

Figure 2: The repairing performance of AugMix [10], DeepRepair [41], ArchRepair [26] and our MetaRepair for ConvNeXt on the representative corruptions for both CIFAR-10 and *tiny*-ImageNet.

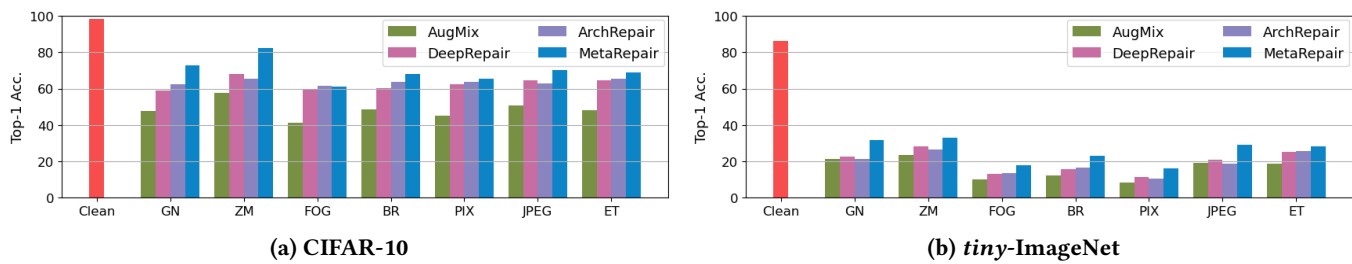

(a) CIFAR-10

(b) *tiny*-ImageNet

Figure 3: The repairing performance of AugMix [10], DeepRepair [41], ArchRepair [26] and our MetaRepair for VAN on the representative corruptions for both CIFAR-10 and *tiny*-ImageNet.

the clean training split data and the model with the best evaluation results on the testing split are saved as the DNN for deployment.

*5.1.5 Baselines.* We collect state-of-the-art (SOTA) repairing results of DenseNet-121 [12] on CIFAR-10 [16] and *tiny*-ImageNet [18]. Specifically, we compare the repairing results of DenseNet-121 with Apricot [42], AugMix [10], Arachne [30], SENSEI [6], DeepRepair [41], ArchRepair [26] on both two datasets. As for the repairing performance of ConvNext [25] and VAN [8], we only compare their repairing results with the best three based on the results for DenseNet which are AugMix [10], DeepRepair [41] and ArchRepair [26]. For all the results, we also show the DNN performance on clean data without any corruption for better comparison.

## 5.2 How MetaRepair performs compared with state-of-the-art DNN repairing approaches?

We first compare our MetaRepair with other repairing approaches. Tab. 1 and Tab. 2 show the repairing results of DenseNet for target corruptions on CIFAR-10 and *tiny*-ImageNet dataset respectively. It is clear that our approach outperforms all the other repairing approaches by a large margin. Notably, MetaRepair achieves 89.45% top-1 accuracy for zoom blur on CIFAR-10, which is only 6.24% lower compared with the performance on clean data. Similarly, it realizes 12.0% accuracy improvement for gaussian noise compared with DeepRepair. While CIFAR-10 is a quite easy dataset, MetaRepair still establish a new repair baseline on *tiny*-ImageNet as shown in Tab. 2. In summary, our proposed repairing method consistently better repairing DenseNet under different corruptions on both datasets.

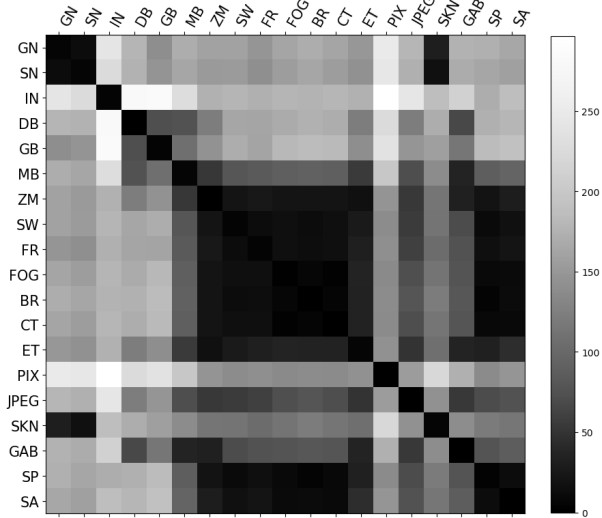

Figure 4: Cross-corruption FID scores based on the Inception v3 [36] features of collected failures for each corruption on CIFAR-10 dataset.

As shown in Fig. 2 and Fig. 3, both ConvNeXt and VAN also receive great performance boosting for different target corruptions on CIFAR-10 dataset. While the repairing performance on *tiny*-ImageNet looks discouraging compared with clean results, our MetaRepair still generally mitigated the performance gap between the clean and corrupted data. Overall, our proposed repairing method realized superior performance improvements which are consistent across different corruptions, datasets and DNNs.

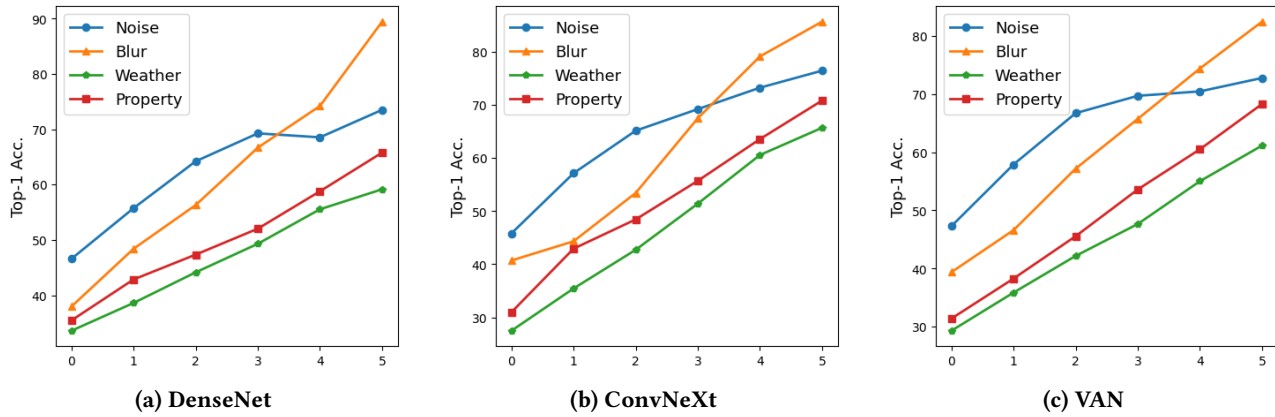

**Figure 5: The repairing performance by gradually adding correlated corruptions as repairing experiences on CIFAR-10 dataset.**

## 5.3 Does the hypothesis in Sec. 4.1 holds for DNN repairing problem?

Although the performance boosting looks promising, our strategy is based on the critical hypothesis from Sec. 4.1. As a general demonstration of its genuineness, we first calculate the cross-corruption FID scores based on the collected failures for all the corruptions on CIFAR-10. As illustrated in Fig. 4, it is clear that different corruptions are correlated on their data distributions which clearly support the hypothesis with evidence. Moreover, we conduct experiments to further verify the coaction of correlated corruptions during repairing. Specifically, we gradually add five most correlated corruptions as experiences to repair the selected target corruption as follows.

- Gaussian Noise: +Shot Noise, +Speckle Noise, +Glass Blur, +Elastic Transform, + Frost
- Zoom Blur: +Elastic Transform, +Spatter, +Glass Blur, +Contrast, +Fog
- Fog: +Saturate, +Spatter, +Contrast, +Brightness, +Frost
- Brightness: +Spatter, +Contrast, +Saturate, +Fog, +Snow

As shown in Fig. 5, it can be seen that the repairing performance of all three DNNs are consistently improved with the adding of correlated corruptions. Especially for blur, weather and image property corruptions, the performances are improved monotonically with the most correlated corruption added. Particularly, we notice the performance boosting of noise corruptions degenerated when the last three corruptions are added. Such trending is correspond to the FID correlations shown in Fig. 4 where only SN and SKN are highly correlated with GN.

## 5.4 Can the DNN generalize to different deploying environments after repair?

One of the main goal with our L2R strategy is to repair DNN with generalizability. In this section, we conduct experiments for verifying the generalization of our repairing strategy. In more details, we consider two types of settings:

- **Basic Generalization** Intuitively, we expect the DNN repaired on specific target corruption can also perform correctly on the corruptions of the same type. For example, the DNN repaired on gaussian noise should be more easy to be repaired on other noise, since they share similar underlying distribution. Therefore, we evaluate the DNN on the corruptions of the same type after repairing it for the selected corruption whose result is referred as "*Base*".
- **One-for-all Generalization** Especially, our L2R is capable of repairing DNN for all type of corruptions with one-time learning. Benefit from the designing of meta-learning how to repair, the proposed repairing pipeline is corruption independent. Consequently, our L2R is expect to be generalizable to all types of environmental corruptions when there exists correlated corruptions as experiences.

As shown in Tab. 3, we tabulate the statistics of the basic generalization results for noise corruptions. The results for other type of corruptions can be found in the supplemental material. We first note that the performance of base corruption, *i.e.* Gaussian Noise, degraded a lot compared with the results in Tab. 1 and Tab. 2. It is because, as Fig. 4 indicated, the most correlated corruptions for GN are SN and SKN, both of which have been excluded from the repairing experiences in this experiment. In contrast, the performance of SN and SKN keeps competitive, since GN has been learned by the model which is highly helpful for SN and SKN repairing. Another phenomenon is the inferior performance of IN compared with other noise corruptions, especially on the CIFAR-10 dataset. This can also be explained by the cross-corruption correlations since there is no highly correlated corruption for IN corruption based on Fig. 4. Generally, similar phenomenon can be observed for other three type of corruptions, that proving our approach successfully generalized the repaired DNN to the same type of corruptions for all the three DNNs and two datasets.

As an unique property of our repairing strategy, we demonstrate the one-for-all generalizability in Tab. 4. Basically, we only conduct one-time meta-training on the available corruptions and meta-test for all the selected target corruptions. Surprisingly, even the overall performance has been degraded due to the lack of highly correlated

**Table 3: Basic generalization results for noise corruptions with gaussian noise as repairing basis.**

| Corruption / Model | CIFAR-10 | | | | tiny-ImageNet | | | |
|---|---|---|---|---|---|---|---|---|
| | Base | SN | IN | SKN | Base | SN | IN | SKN |
| DenseNet | 56.76 | 64.52 | 54.39 | 60.40 | 18.45 | 20.05 | 15.30 | 25.09 |
| ConvNeXt | 58.49 | 66.27 | 55.46 | 62.30 | 18.38 | 22.83 | 15.33 | 20.29 |
| VAN | 55.19 | 61.32 | 51.77 | 61.89 | 25.40 | 29.93 | 18.58 | 29.18 |

**Table 4: One-for-all generalization results for different type of corruptions on CIFAR-10 and tiny-ImageNet datasets.**

| | Model | GN | ZM | FOG | BR | PIX | JPEG | ET |
|---|---|---|---|---|---|---|---|---|
| CIFAR-10 | DenseNet | 64.39 | 65.29 | 54.48 | 52.33 | 60.49 | 64.62 | 54.15 |
| | ConvNeXt | 65.46 | 64.96 | 53.23 | 58.70 | 59.10 | 62.04 | 56.68 |
| | VAN | 61.77 | 65.54 | 50.83 | 55.73 | 61.46 | 64.58 | 58.03 |
| tiny-ImageNet | DenseNet | 20.05 | 13.74 | 15.26 | 18.82 | 19.33 | 21.79 | 14.30 |
| | ConvNeXt | 22.83 | 24.40 | 18.16 | 16.73 | 25.38 | 23.52 | 21.80 |
| | VAN | 29.93 | 26.77 | 13.20 | 16.80 | 15.72 | 23.21 | 24.07 |

**Table 5: Ablation study results of MetaRepair on CIFAR-10 dataset.**

| | Corruption | Empirical Selection | w/o Confidence | Separate Update | MetaRepair |
|---|---|---|---|---|---|
| DenseNet | GN | 69.60 | 58.07 | 73.20 | **73.51** |
| | ZM | 75.86 | 79.05 | 85.22 | **89.45** |
| | FOG | 55.17 | 53.76 | 58.09 | **59.15** |
| | BR | 61.85 | 60.46 | 63.81 | **65.78** |
| ConvNeXt | GN | 68.48 | 63.60 | 75.11 | **76.46** |
| | ZM | 61.50 | 62.98 | 68.54 | **71.69** |
| | FOG | 55.45 | 58.54 | 64.97 | **65.70** |
| | BR | 60.98 | 64.67 | 70.98 | **70.99** |
| VAN | GN | 64.72 | 56.40 | 71.00 | **72.77** |
| | ZM | 66.86 | 61.38 | 67.29 | **70.45** |
| | FOG | 53.12 | 53.15 | 60.21 | **61.15** |
| | BR | 63.71 | 60.62 | 64.31 | **68.30** |

repairing experiences, our approach still achieves promising results on all the target corruptions. Similar to the basic generalization results, we can observe the dominant influence of the corruption correlations during repairing. For example, there is no significant performance degradation for PIX compared with the best results in Tab. 1, since all other corruptions are not highly correlated with it. In summary, L2R exhibit superior generalizability for repairing DNN under different type of corruptions which is highly desirable for deploying DNN for real environments.

## 5.5 What are the contributions of different MetaRepair components?

In this part, we ablate different versions of the designed MetaRepair on CIFAR-10 dataset. In specific, we experiment three settings as follows: ❶ We first test whether selecting corruptions with FID as repairing experience is more effective compared with empirical selection. This setting is denoted as "*Empirical Selection*". ❷ Then we utilize the corruptions selected with FID score to test whether the confidence of selected experiences influence the overall repairing performance, which is denoted as "*w/o Confidence*". ❸ With the optimal experience selection strategy, we finally test the influence of accumulated updates for the tasks in an testing episode. We denote this setting as "*Separate Update*" which means DNN is updated with tasks in separate.

As Tab. 5 shows, we first observe that separately update DNN with tasks during meta-test has limiting impact to the overall performance. We attribute this to the goal divergence between adapting to data and adapting to corruption, while accumulate updating still benefit to the overall results in general. In contrast, it is clear both selection with FID and confidence contribute to the repairing performance a lot. Such results conform to the cross-corruption correlation we have concluded in Sec. 5.3. We also notice that the "w/o Confidence" performance of GN drops a lot when compare with other corruptions, which can be ascribed to the disturbance of unrelated corruptions.

## 6 CONCLUSION

In this work, we propose to repair DNN from a novel perspective by learning how the DNN is repaired under different deploying environments. Instead of focusing on rectifying DNN misbehaviour for specific corruption, the proposed Learning to Repair (L2R) strategy formalize DNN repair into a meta-learning problem to acquire the general repairing capability. By instantiate L2R with DNN repair specific considerations, the designed MetaRepair model effectively boosts the repairing performance and realizes great repairing generalization on various type of corruptions. Practically, L2R simply abstracts different repairing experiences and applies the meta-learned repairing capability, which is flexible and general for implementation. The comprehensive experiments confirmed the consistent superiority of the proposed strategy and corresponding MetaRepair model for three modern DNNs on two datasets. As potential future works, the proposed MetaRepair will be implemented with different meta-learning algorithms and evaluated on larger benchmarks for comprehensive understanding of the learning to repair strategy.

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
