# OpenReview forum: "MetaRepair: Learning to Repair Deep Neural Networks from Repairing Experiences"
_acmmm.org/ACMMM/2024/Conference — MM2024 Poster_

### Official Review · Reviewer_98fi · 2024-04-30

**Rating:** 5
**Confidence:** 2

**Summary:**

The authors introduce MetaRepair, a novel method designed to enhance the generalization of repair DNN, and evaluate its effectiveness across various models and datasets.

**Strengths:**

1. The perspective is novel and interesting. The author combines mete-learning with repair DNN to improve its generalization ability.

2. The methodology is clear and straightforward, and the extensive experimental results effectively validate the approach.

3. The experimental results convincingly validate the approach.

4. The paper is overall well-written and easy to follow.

**Limitations:**

While the current version of the paper is satisfactory, addressing the following points would still be meaningful.

1. The specific task-related loss function utilized in this research is not clearly defined.

2. The paper should discuss the additional computational overhead introduced by incorporating meta-learning.

**Suitability:**

2

---

### Official Review · Reviewer_iTVX · 2024-05-21

**Rating:** 4
**Confidence:** 1

**Summary:**

This work proposes to repair DNN from a novel perspective, i.e. Learning to Repair (L2R), where the repairing of target DNN is realized as a general learning-to-learn, a.k.a. meta-learning, process. In specific, observing different corruptions are correlated on their data distributions, this paper proposes to utilize previous DNN repair experiences as tasks for meta-learning how to repair the target corruption

**Strengths:**

(1) This paper provides a novel perspective for DNN repair and redefines the conventional DNN repairing into a metalearning process and elaborate the new challenges as well as the model, dataset and learning setup.

(2) This paper proposes a novel Learning to Repair (L2R) strategy to repair DNN with generalization.

**Limitations:**

(1) This paper only considers two simple datasets, CIFAR-10 and Tiny-ImageNet. More complex datasets, such as ImageNet, should also be considered.

(2) The paper uses baseline models from studies published before 2022. More advanced baselines should be included.

**Suitability:**

1

---

### Official Review · Reviewer_GMhm · 2024-05-24

**Rating:** 3
**Confidence:** 2

**Summary:**

The paper proposes MetaRepair for repairing deep neural networks by employing meta-learning, aiming to enhance generalization under specific environmental corruptions; it also seems to be promising in repairing DNNs across different corruptions.

**Strengths:**

1.Leveraging meta-learning to acquire knowledge from repairing experiences is intriguing and presents a feasible approach to enhancing the robustness of deep neural networks in challenging scenarios.

2.The experimental results seem to be competitive and demonstrate the efficacy of MetaRepair in addressing various forms of corruption.

3.The ability of MetaRepair to generalize across different corruption scenarios, when initially tailored to a specific type of corruption, enhances its practical utility in real-world settings.

**Limitations:**

1.The successful implementation of MetaRepair requires precise identification of a type of corruption under deployment, which in the experiments appears to be assumed as prior knowledge. However, this assumption is not trivial, especially in real-world scenarios where new samples arrive in a streaming fashion. How can one determine if a new sample is corrupted and, if so, which type of corruption it belongs to? Given that each target corruption necessitates a separate meta-learning process, deploying a single DNN could potentially evolve into a mixture of experts. In this model, a gating mechanism would route specific inputs to the expert model trained to handle that particular corruption. This approach, however, introduces a risk of misrouting inputs to incorrect expert models, an error which should be considered in the overall evaluation of the system's robustness and reliability.

2.The process of collecting repair experiences and preparing target corruption training data appears to necessitate expert knowledge for identifying and clustering corruptions, clearly distinguishing each exploitable type. This assumption is quite strong, and there is also potential label noise. Furthermore, in scenarios where sufficient expert knowledge is available to clearly identify the target corruption, whether it is feasible to directly model this corruption through data augmentation? For instance, adding Gaussian noise to original data could be a direct and efficient strategy to specifically address Gaussian corruption.

3.The assumptions under domain/out-of-distribution generalization, domain adaptation, and repair of corruption are distinctly different. The assumption for repair seems particularly strong, requiring data from multiple source domains and clear knowledge of the new corruption to be addressed. Although MetaRepair demonstrates some ability to generalize across different corruptions, whether it remains sufficiently effective when compared fairly across different knowledge settings. For instance, AugMix appears capable of achieving single domain generalization without relying on any additional domain data. Is a comparison between MetaRepair and such methods fair, considering MetaRepair's dependency on specific prior knowledge about the corruption types?

I am not an expert in DNN repair and do not have sufficient confidence in my own judgments. I am open to the authors' responses and further discussion.

**Suitability:**

2

---

### Meta-Review · Area_Chair_9X2q · 2024-06-30

**Recommendation:** Accept (Poster)
**Confidence:** 3

**Metareview:**

This paper proposes a novel method for repairing deep neural networks. The repairing of target DNN is realized as a learning-to-learn framework. This paper receives mixed final ratings: two accepts and one borderline reject. After checking the paper, reviews, and rebuttal document, I tend to accept this paper.

The AC thanks Reviewer GMhm's efforts in reviewing this paper. It seems that Reviewer GMhm is not an expert in this area, and hasn't published many related papers in this field. Reviewer GMhm also pointed out this in their review. Therefore, I have to downgrade the importance of Reviewer GMhm's review and give the final recommendation mainly based on other reviewers' comments.